

# *Rhopalocnemis phalloides* has one of the most reduced and mutated plastid genomes known

Mikhail I. Schelkunov[1,2], Maxim S. Nuraliev[3,4] and
Maria D. Logacheva[1,5]

[1] Skolkovo Institute of Science and Technology, Moscow, Russia
[2] Institute for Information Transmission Problems, Moscow, Russia
[3] Faculty of Biology, Moscow State University, Moscow, Russia
[4] Joint Russian–Vietnamese Tropical Scientific and Technological Center, Cau Giay,
Hanoi, Vietnam
[5] A.N. Belozersky Research Institute of Physico-Chemical Biology, Moscow State University,
Moscow, Russia

Corresponding author
Mikhail I. Schelkunov,
shelkmike@gmail.com

## ABSTRACT

Although most plant species are photosynthetic, several hundred species have lost the
ability to photosynthesize and instead obtain nutrients via various types of
heterotrophic feeding. Their plastid genomes markedly differ from the plastid
genomes of photosynthetic plants. In this work, we describe the sequenced plastid
genome of the heterotrophic plant *Rhopalocnemis phalloides*, which belongs to the
family Balanophoraceae and feeds by parasitizing other plants. The genome is highly
reduced (18,622 base pairs vs. approximately 150 kbp in autotrophic plants) and
possesses an extraordinarily high AT content, 86.8%, which is inferior only to AT
contents of plastid genomes of *Balanophora*, a genus from the same family. The gene
content of this genome is quite typical of heterotrophic plants, with all of the
genes related to photosynthesis having been lost. The remaining genes are notably
distorted by a high mutation rate and the aforementioned AT content. The high AT
content has led to sequence convergence between some of the remaining genes and
their homologs from AT-rich plastid genomes of protists. Overall, the plastid genome
of *R. phalloides* is one of the most unusual plastid genomes known.

GC content, AT content

## INTRODUCTION

Though plants are generally considered photosynthetic organisms, there are several
hundred plant species that have lost the ability to photosynthesize during the course of
evolution (*Westwood et al., 2010*; *Merckx et al., 2013*). They feed either by parasitizing
other plants or by obtaining nutrients from fungi. In addition to the completely
heterotrophic plants, there are also plants that combine the ability to photosynthesize with
the heterotrophic lifestyle. They are termed partial heterotrophs (or hemi-heterotrophs,
or mixotrophs) in contrast to the former, which are termed complete heterotrophs
(or holo-heterotrophs).

The completely heterotrophic plants show a high degree of similarity, though there were several dozen cases of independent transition to complete heterotrophy. For example, these plants all either lack leaves or have very reduced leaves. These plants are non-green because of the absence (or at least highly reduced amounts; *Cummings & Welschmeyer, 1998*) of chlorophyll. Additionally, a common feature of many completely heterotrophic angiosperms is that they spend most of their lifetimes underground, since without the need to photosynthesize their only reason to appear aboveground is for flowering and seed dispersal.

Genomic studies of heterotrophic plants are mostly focused on plastid genomes, since (1) most of the plastid genes are related to photosynthesis, and thus changes in the plastid genomes are expected to be more prominent compared to mitochondrial and nuclear genomes, and (2) plastid genomes are smaller than nuclear and mitochondrial ones and usually have higher copy numbers and are thus easier to sequence (*Daniell et al., 2016*; *Gualberto & Newton, 2017*; *Sakamoto & Takami, 2018*). The main feature of the plastid genomes of complete heterotrophs is the loss of genes responsible for photosynthesis and respective shortening of the genomes, from approximately 150 kbp (typical of autotrophic plants) to, in the most extreme known case, 12 kbp (*Bellot & Renner, 2015*; *Graham, Lam & Merckx, 2017*; *Wicke & Naumann, 2018*). The remaining genes are the ones with functions not related to photosynthesis. Usually they are *accD* (a gene whose product participates in fatty acid synthesis—one of the plastid functions besides photosynthesis), *clpP* (encodes a component of a complex responsible for degradation of waste proteins in plastids), *ycf1* (thought to encode a component of the translocon—the complex which imports proteins from cytoplasm into plastids), *ycf2* (a conserved gene present in almost all plants, but with unknown function) and various genes required for translation of the aforementioned ones, namely genes that code for protein and RNA components of the plastid ribosome and for tRNAs. One of the tRNA-coding genes, *trnE-UUC*, also has an additional function, with its product participating in haem synthesis (*Kumar et al., 1996*).

In addition to the expected shortening of the genome, there are some peculiar and still unexplained features in the plastid genomes of heterotrophic plants, namely their increased mutation accumulation rate (*Bromham, Cowman & Lanfear, 2013*; *Wicke & Naumann, 2018*) and increased AT content (*Wicke & Naumann, 2018*). In the most extreme cases, plastid genomes of heterotrophic plants may accumulate mutations approximately 100 times faster than their closest autotrophic relatives (*Bellot & Renner, 2015*). The most obvious explanation, the relaxation of selection, is refuted by the fact that dN/dS (a common measure of selective pressure) is usually not increased in the plastid genes of heterotrophic plants, except for photosynthesis-related genes during their pseudogenization, but the mutation accumulation rate is high even after the loss of all such genes (*Logacheva, Schelkunov & Penin, 2011*; *Barrett et al., 2014*; *Schelkunov et al., 2015*; *Lam, Soto Gomez & Graham, 2015*; *Wicke et al., 2016*; *Naumann et al., 2016*). AT content is increased from approximately 65% in autotrophic species (*Smith, 2012*) to 88.4% in the most prominent case among heterotrophic species (*Su et al., 2019*), also because of an unknown reason.

Genes not related to photosynthesis, such as *accD* and *infA*, are sometimes transferred to the nuclear genome (*Millen et al., 2001*; *Rousseau-Gueutin et al., 2013*; *Liu et al., 2016*). Therefore, when all genes with functions not related to translation are transferred to the nuclear genome, there will be no reasons to keep the translation apparatus in plastids, and the genes responsible for translation will also be lost. Thus, the plastid genome is potentially able to disappear entirely. Indeed, two putative cases of the complete plastid genome loss are known: one in algae of the genus *Polytomella* (*Smith & Lee, 2014*) and the other one in the parasitic plant *Rafflesia lagascae* (*Molina et al., 2014*); the second case is disputable (*Krause, 2015*).

The initial aim of the present study was to prove that the completely heterotrophic plant *Rhopalocnemis phalloides* had also lost its plastid genome completely. *Rhopalocnemis phalloides* is a parasitic plant from the family Balanophoraceae (order Santalales) which occurs in Asia and feeds by obtaining nutrients from roots of various plants. Initially, we sequenced approximately 10 million pairs of reads on the HiSeq 2000 platform and observed no contigs with similarity to typical plastid genes, while there were obvious mitochondrial contigs. Based on our experience in studying plastid genomes of heterotrophic plants, mitochondrial contigs usually have lower sequencing coverage than plastid contigs; thus, the plastid genome is always easier to assemble. This led us to suppose that the plastid genome in *Rhopalocnemis phalloides* may have been completely lost. To verify this, we sequenced approximately 200 million pairs of additional reads. What we found is that the plastid genome *is* in fact present, but its tremendous AT content (86.8%) hampered PCR, which is one of the usual steps in library preparation of Illumina, and thus the sequencing coverage of the genome was much lower than one might have expected. This article is dedicated to the analysis of this plastid genome.

## MATERIALS AND METHODS

### Sample collection and sequencing

The specimen of *Rhopalocnemis phalloides* was collected during an expedition of the Russian-Vietnamese Tropical Centre in Kon Tum Province, Vietnam, in May 2015 (voucher information: Southern Vietnam, Kon Tum Prov., Kon Plong distr., Thach Nham protected forest, 17 km N of Mang Den town, in open forest, 14°45′15″N 108°17′40″E, elev. 1,400 m, Nuraliev M.S., Kuznetsov A.N., Kuznetsova S.P., No. 1387, April 18, 2015). The studied material was preserved in silica gel and in RNAlater. The voucher is deposited at the Moscow University Herbarium (MW) (*Seregin, 2018*) with the barcode MW0755444.

DNA was extracted from an inflorescence using a CTAB-based method (*Doyle, 1987*), and the DNA library was prepared using the NEBNext DNA Ultra II kit (New England Biolabs, Ipswich, MA, USA). Sequencing was performed with a NextSeq 500 sequencing machine (Illumina, San Diego, CA, USA) in the paired end mode, producing 387,351,294 reads (193,675,647 read pairs), each 150 bp long.

RNA was extracted from an inflorescence using the RNeasy Mini kit (Qiagen, Hilden, Germany). Plastid transcripts are usually not polyadenylated, so the method of poly(A) RNA selection was not applicable in our study. Instead, we used a protocol based on

depletion of ribosomal RNA with the Plant Leaf Ribo Zero kit (Illumina, San Diego, CA, USA). The RNA-seq library was prepared using the NEBNext RNA Ultra II kit (New England Biolabs, Ipswich, MA, USA) and sequenced on a HiSeq 2500 sequencing machine (Illumina, San Diego, CA, USA) with TruSeq reagents v.4 in the paired end mode, producing 54,794,466 reads (27,397,233 read pairs), 125 bp each.

## Genome assembly and annotation

Both DNA-seq and RNA-seq reads were trimmed by Trimmomatic 0.32 (*Bolger, Lohse & Usadel, 2014*) in the palindromic mode, removing bases with quality less than 3 from the 3′ ends of reads, and fragments starting from 4-base-long windows with average quality less than 15 (SLIDINGWINDOW:4:15). Reads that, after trimming, had average quality less than 20 or length shorter than 30 bases were removed.

The assembly was performed from DNA-seq reads by two tools. First, it was made by CLC Assembly Cell 4.2 (https://www.qiagenbioinformatics.com/products/clc-assembly-cell/) with the default parameters. Second, it was made by Spades 3.9.0 (*Bankevich et al., 2012*). Because the performance of Spades is slow when running on large number of reads, prior to starting its assembly we removed from reads k-mers with coverage less than 50× by Kmernator 1.2.0 (https://github.com/JGI-Bioinformatics/Kmernator). This allowed us to eliminate most reads belonging to the nuclear genome (and, potentially, some reads belonging to low-covered plastid regions), thus highly reducing the number of reads. The Spades assembly was run on this reduced read set, with the "–only-assembler" and "–careful" options. To determine the read coverage of contigs in these two assemblies, we aligned to them reads by CLC Assembly Cell 4.2, requiring at least 80% of the length of each read to align with a sequence similarity of at least 98%.

To find contigs potentially belonging to plastid and mitochondrial genomes, we aligned by BLASTN and TBLASTN from BLAST 2.3.0+ suit (*Camacho et al., 2009*) proteins and non-coding RNA (ncRNA) genes from reference species. As the references, we used sequences from the plastid genomes of *Balanophora reflexa* (NCBI accession KX784266), *B. laxiflora* (NCBI accession KX784265), *Viscum album* (NCBI accession NC_028012), *Osyris alba* (NCBI accession NC_027960), *Arabidopsis thaliana* (NCBI accession NC_000932), *Nicotiana tabacum* (NCBI accession NC_018041) and mitochondrial genomes of *V. album* (NCBI accession NC_029039), *Citrullus lanatus* (NCBI accession GQ856147), *Mimulus guttatus* (NCBI accession NC_018041). *Balanophora*, *V. album* and *O. alba* were used because they, like *Rhopalocnemis phalloides*, belong to Santalales. Other species were chosen because they belong to other orders of eudicots. Alignment was performed with the maximum $e$-value of $10^{-3}$ and low complexity filter switched off. The word size was 7 for BLASTN and 3 for TBLASTN. Here and later, the local BLAST was used with the parameter "max_target_seqs" set to $10^9$ to avoid the problem discussed by *Shah et al. (2018)*, who state that BLAST results may be improper when this parameter is set to a small value.

Five contigs containing plastid genes were found in the CLC assembly and three contigs in the Spades assembly. After aligning contigs of these two assemblies to each other (BLASTN, maximum $e$-value $10^{-3}$, word size 7, low complexity filter switched off), it appeared that the places at which the CLC contigs were broken by gaps corresponded to

continuous places in the Spades contigs and, vice versa, gaps in the Spades contigs corresponded to continuous places in the CLC contigs. This allowed us, by joining the contigs of these two assemblies, to create a circular sequence corresponding to the plastid genome. To check the assembly, we mapped reads (in CLC Assembly Cell 4.2, requiring at least 80% of the length of each read to align with a sequence similarity of at least 98%) to the resultant sequence and verified (by eye, in CLC Genomics Workbench 7.5.1, https://www.qiagenbioinformatics.com/products/clc-genomics-workbench/) that there were no places uncovered by reads and no places where the insert size abruptly decreased or increased. Such places of abrupt increase or decrease of the insert size may indicate regions with assembly errors, consisting of sequence insertions or deletions, respectively. As read mapping is complicated on the edges of a sequence, we also performed such analysis on a reoriented version of the plastid genome, in which the sequence was broken in the middle and the ends were joined. These analyses indicated that the assembly contained no errors.

To find genes in the plastid genome, we used a complex strategy, because highly mutated genes may be hard to notice. We used the following methods:

1. The alignment of reference protein-coding and ncRNA-coding genes by BLASTN and TBLASTN, as described above.

2. Open reading frames were scanned by InterProScan 5.11 (*Jones et al., 2014*) using the InterPro 51.0 (*Finn et al., 2017*) database with the default parameters. "Open reading frames" here were any sequences at least 20 codons long uninterrupted by stop codons. Not requiring an ORF to begin from a start-codon allowed for the detection of exons in multi-exonic genes.

3. The genome was scanned by Infernal 1.1.2 (*Nawrocki & Eddy, 2013*) with RNA models from Rfam 12.2 database (*Nawrocki et al., 2015*) to predict ncRNA-coding genes. The maximum allowed $e$-value was set to $10^{-3}$.

4. To predict rRNA-coding genes, RNAmmer 1.2 server (*Lagesen et al., 2007*) was used in bacterial mode and eukaryotic mode.

5. The genome was scanned by tRNAscan-SE 1.23 (*Lowe & Eddy, 1997*) with the default parameters, in the organellar (models trained on plastid and mitochondrial tRNAs) and also in the general (models trained on tRNAs from all three genomes) mode, to predict tRNA-coding genes.

6. The genome was annotated by DOGMA (*Wyman, Jansen & Boore, 2004*) and Verdant (*McKain et al., 2017*).

7. When determining which ATG codon was a true start codon, we compared the sequence of a gene with sequences of its homologs from the aforementioned reference species.

8. To determine exon borders, RNA-seq reads with a minimum length of 100 bp (to minimize false mappings) were mapped to the genome by CLC Assembly Cell 4.2, requiring at least 50% of each read's length to map with a sequence similarity of at least 90%. Exon borders were found by eye in CLC Genomics Workbench 7.5.1 as regions of genes in which there were many partially mapped reads. The exon borders of the reference species were used for comparison.

9. To check for RNA editing that could create new start or stop codons, we mapped RNA-seq reads with a minimum length of 100 bp by CLC Assembly Cell 4.2, requiring at least 80% of each read's length to map with a sequence similarity of at least 90%. Mismatches between the reads and the genome were inspected by eye in CLC Genomics Workbench 7.5.1.

10. After annotating the genes, we additionally verified that there were no remaining regions with high sequence complexity, relatively low AT content or high coverage by RNA-seq reads where no genes were predicted. Regions with high sequence complexity were predicted in the genome by CLC Genomics Workbench 7.5.1 using K2 algorithm (*Wootton & Federhen, 1993*) with a window size of 101 bp. The AT content plot was created by a custom script with 200-bp-long windows. RNA-seq reads with a minimum length of 100 bp were mapped by CLC Assembly Cell 4.2, requiring at least 80% of each read's length to map with a sequence similarity of at least 90%.

After completing gene prediction, the plastid genome was reoriented to start from the first position of *rps14*, as this is the first gene in the canonical representation of the plastid genome of *Arabidopsis thaliana* which is also present in the plastid genome of *Rhopalocnemis phalloides*.

## Estimation of contamination amount

The nuclear genome size could be overestimated if, in addition to the own DNA of *Rhopalocnemis phalloides*, contaminating DNA was sequenced. For example, this contamination may originate from endophytic bacteria and fungi. To estimate the amount of contamination, 1,000 random DNA-seq read pairs, taken after the trimming, were aligned by BLAST to NCBI databases. Taxonomies of their best matches were used as proxies for the reads' source taxonomies. To increase the sensitivity of the search, the analysis was performed as follows:

1. All reads were aligned to NCBI NT (the database current as of September 18, 2017) by BLASTN from BLAST 2.3.0+ suite with the maximum allowed $e$-value of $10^{-3}$ and the word size of seven bp. To decrease the number of false-positive matches, hard masking of low-complexity regions ("–soft_masking false" option) was used.

2. All reads were aligned to NCBI NR (the database current as of September 18, 2017) by BLASTX from BLAST 2.3.0+ suite with the maximum allowed $e$-value of $10^{-3}$ and the word size of three bp. Hard masking in BLASTX is enabled by default.

3. If at least one of two reads in a pair had matches to NT, the taxonomy of the match with the lowest $e$-value was considered the taxonomy of the read pair. If the read pair had no matches in NT, the taxonomy of the match to NR with the lowest $e$-value was considered the taxonomy of the read. Therefore, the alignment to NT had higher priority than the alignment to NR. This was done to take into account synonymous positions of genes, where possible, and thus increase the precision of the taxonomic assignment of read pairs.

## Other analyses

To determine the phylogenetic placement of *Rhopalocnemis phalloides* within Balanophoraceae, we utilized the alignment of genes from 186 species (180 species of Santalales plus 6 outgroup species) created by *Su et al. (2015)*. *Rhopalocnemis phalloides* was not studied in that article. Seven genes were used for the phylogenetic analysis in that work: plastid *accD*, *matK*, *rbcL*; nuclear 18S rDNA, 26S (also known as 25S) rDNA and *RPB2*; and mitochondrial *matR*. As *matK* and *rbcL* are absent from the plastid genome of *Rhopalocnemis phalloides*, we were unable to use them. *accD* of *Rhopalocnemis phalloides* contains many mutations and thus can be aligned improperly, so we did not use it either. Mitochondrial *matR* is disrupted in *Rhopalocnemis phalloides* by several frameshifting indels. Owing to the large size of the nuclear genome of *Rhopalocnemis phalloides* (see the paragraph "Other genomes of *Rhopalocnemis phalloides*"), *RPB2* had a low coverage, and its sequence could not be obtained from the available DNA-seq reads. The sequences of 18S rDNA and 26S rDNA were easier to determine, as they had many copies in the nuclear genome and thus their coverage was higher. To find their sequences among the contigs, we aligned 18S rDNA and 26S rDNA of *Arabidopsis thaliana* by BLASTN with the default parameters to the contigs of the Spades assembly. The sequences of 18S rDNA and 26S rDNA were added to the alignment of *Su et al. (2015)* using MAFFT 7.402 (*Katoh & Standley, 2013*) with options addfragments and maxiterate 1000. The phylogenetic tree was built with RAxML 8.2.4 (*Stamatakis, 2014*), utilizing 20 starting stepwise-addition parsimony trees, employing GTR+Gamma model, with the same six outgroup species as in the work of *Su et al. (2015)* (*Antirrhinum majus*, *Arabidopsis thaliana*, *Camellia japonica*, *Cornus florida*, *Myrtus communis* and *Spinacia oleracea*). The required number of bootstrap pseudoreplicates was determined by RAxML automatically with the extended majorityrule consensus tree criterion ("autoMRE"). The tree was visualized with FigTree 1.4.3 (http://tree.bio.ed.ac.uk/software/figtree/).

To compare the substitution rate in the plastid genome of *Rhopalocnemis phalloides* with substitution rates in other species of Santalales, we used common protein-coding genes of *Rhopalocnemis phalloides*, *B. reflexa*, *Arabidopsis thaliana* (used as the outgroup) and the only four species of Santalales with published plastid genomes as of 2017: *V. album*, *O. alba*, *Champereia manillana* and *Schoepfia jasminodora*. Their protein-coding gene alignment was created by TranslatorX 1.1 (*Abascal, Zardoya & Telford, 2010*) based on an alignment of the corresponding amino acid sequences performed by Muscle 3.8.31 (*Edgar, 2004*) with the default parameters. *ycf1*, *ycf2* and *accD* of *Rhopalocnemis phalloides* differed from the homologous genes of other species so much that a reliable alignment was not possible. Alignments of other genes were then concatenated into a single alignment and provided to Gblocks server (*Castresana, 2000*), which removed poorly aligned regions from the alignment. Gblocks was run in the codon mode, with the default parameters. Substitution rates and selective pressure were evaluated by codeml from PAML 4.7 (*Yang, 2007*) with the F3×4 codon model, starting dN/dS value of 0.5 and starting transition/transversion rate of 2. The phylogenetic tree provided to PAML was a subtree of the large phylogenetic tree of Santalales, produced as described above.

Additionally, the analysis of substitution rates and selective pressure was performed by BppSuite 2.3.2 (*Guéguen et al., 2013*). To the best of our knowledge, this is the only tool that is capable of phylogenetic analyses of protein-coding sequences that takes into account different codon frequencies in different sequences (*Guéguen & Duret, 2017*), whereas PAML uses a single averaged codon frequency for all sequences. This is important because the codon frequencies in *Rhopalocnemis phalloides* and *B. reflexa* highly differ from the codon frequencies in the mixotrophic Santalales of comparison. The program bppml from BppSuite was run using a nonhomogeneous ("one_per_branch") model, the substitution model was YN98, the codon model F3×4, starting dN/dS values of 0.5 and starting transition/transversion rates of 2. Starting branch lengths were 0.1 substitutions per codon. The parameter estimation was performed by the full-derivatives method with optimization by the Newton–Raphson method ("optimization=FullD(derivatives=Newton)"), using parameters transformation ("optimization.reparametrization=yes").

To check for similarity between the genes and the proteins of the *Rhopalocnemis phalloides* plastid genome and sequences from other species, we performed BLASTN and BLASTP alignment against NCBI NT and NR databases, respectively, on the NCBI website (https://blast.ncbi.nlm.nih.gov/Blast.cgi) on March 4, 2018 with the default parameters.

To build the phylogenetic tree of *rrn16*, we took sequences of *rrn16* from plastid genomes of Embryophyta and SAR ("Stramenopiles, Alveolata, Rhizaria," a clade of protists), one random representative per order. The plastid genomes were taken from the RefSeq database (current as of May 12, 2019), the taxonomy from NCBI Taxonomy (current as of May 13, 2019). Overall, there were 93 representatives of Embryophyta and 35 representatives of SAR. To these sequences we added the sequences of *rrn16* from *Rhopalocnemis phalloides*, *Corynaea crassa* (NCBI accession U67744), *B. japonica* (NCBI accession KC588390), *Nitzschia* sp. IriIs04 (NCBI accession AB899709) and *Plasmodium cynomolgi* (NCBI accession AB471804). These particular sequences of *Nitzschia* and *Plasmodium* were randomly chosen among the SAR sequences that produced matches to the *rrn16* of *Rhopalocnemis phalloides* in the BLAST search described in the previous paragraph. There were also matches to *Leucocytozoon* in that analysis, but *Leucocytozoon caulleryi* had already been randomly chosen as a representative for the SAR order Haemosporida, so there was no need to add it again. The sequences of these 133 *rrn16* genes were aligned by MAFFT 7.402 with the option—maxiterate 1000. Poorly aligned regions were removed by Gblocks server with the option "Allow gap positions within the final blocks" switched on. Without this option Gblocks is too severe for alignments which contain many sequences. The unrooted phylogenetic tree was built by RAxML 8.2.4 with the parameters described above. Branches corresponding to clades that had bootstrap support values below 70 were collapsed by TreeGraph 2.14.0-771 beta (*Stöver & Müller, 2010*). The tree was visualized by FigTree 1.4.3.

Codon usage and amino acid usage of the common protein-coding genes of *Rhopalocnemis phalloides* and species of comparison were calculated by CodonW 1.4.2 (*Peden, 1999*). Frequencies of 21-bp-long k-mers were calculated for the trimmed DNA-seq reads by Jellyfish 2.1.2 (*Marçais & Kingsford, 2011*), not using the Bloom filter (to count the number of low-frequency k-mers precisely).

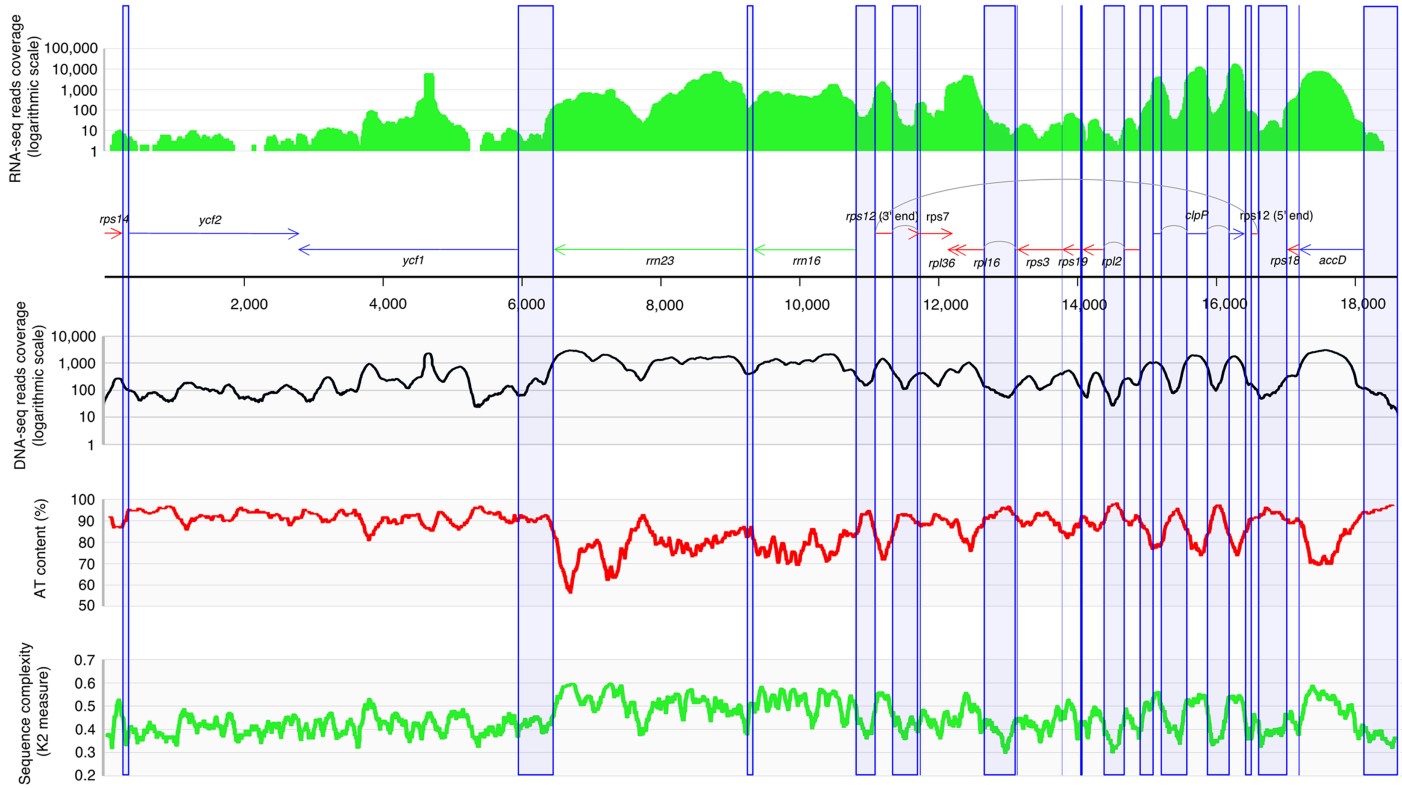

**Figure 1 Map of the *Rhopalocnemis phalloides* plastid genome showing various features.** The circular-mapping plastid genome is represented linearly for convenience. Green arrows are rRNA-coding genes, red arrows are ribosomal protein-coding genes and blue arrows are genes coding proteins with other functions. Gray arcs represent splicing. Blue columns show non-coding regions.

The list of plastid genomes with their lengths and AT contents was obtained from the NCBI database (https://www.ncbi.nlm.nih.gov/genome/browse/#!/organelles/). Information on whether a specific plant species is completely heterotrophic was obtained by literature analysis.

## RESULTS AND DISCUSSION

### The gene content of the *Rhopalocnemis phalloides* plastid genome

The plastid genome of *Rhopalocnemis phalloides* is circular-mapping and has a length of 18,622 bp long. Its map is represented in Fig. 1, in a linear form, for convenience. The protein-coding gene content is quite typical for highly reduced plastid genomes of completely heterotrophic plants (*Graham, Lam & Merckx, 2017*; *Wicke & Naumann, 2018*). The plastid genome of *Rhopalocnemis phalloides* possesses the genes *accD*, *clpP*, *ycf1*, *ycf2* (mentioned in the Introduction) and nine genes encoding protein components of the ribosome. Additionally, it codes for *rrn16* and *rrn23*, RNA components of the plastid ribosome. Like several other highly reduced plastid genomes, it lacks *rrn4.5* and *rrn5*—genes coding for two other RNAs of the ribosome—which poses the interesting puzzle of how the ribosome works without these genes in the plastid genome. One possibility is that these genes were transferred either to the mitochondrial or to the nuclear genome

and are now transcribed there and imported to the plastids from the cytoplasm. The other possibility is that the ribosome is capable of working without them, akin to how it can work without some ribosomal proteins (*Tiller & Bock, 2014*). We plan to clarify this question in an upcoming article dedicated to the analysis of the *Rhopalocnemis phalloides* transcriptome.

The tRNA-coding gene content of the *Rhopalocnemis phalloides* plastid genome is also puzzling. The standard method to predict tRNA-coding genes is the program tRNAscan-SE. It has a dedicated "organellar" mode in which tRNA models were trained on mitochondrial- and plastid-encoded tRNA sequences and structures. It also has a "general" mode whose models are based on nuclear-encoded tRNAs. In the organellar mode, the tool predicts 64 tRNA-coding genes, which is much more than the approximately 30 tRNA-coding genes encoded in plastomes of typical autotrophic species (*Wicke et al., 2011*). In the general mode, the tool predicts zero tRNA-coding genes. Our experience in working with different plastid genomes suggests that results of predictions in these two modes usually coincide. Of the 64 predicted tRNA-coding genes, 61 have introns, and the mean AT content of the 64 genes is 94%. Therefore, we supposed that most of them, if not all, were false-positive predictions. They could originate from the ease with which sequences of low complexity form secondary structures—these spuriously generated cloverleaf-like structures may have deceived the algorithms of tRNAscan-SE. Seventeen of the predicted tRNA-coding genes were for isoleucine tRNAs, and 11 were for lysine. This further attested to the false-positive nature of these genes, as false-positively predicted tRNA-coding genes in an AT-rich genome are expected to have AT-rich anticodons, and the anticodons of isoleucine and lysine tRNAs are two of the most AT-rich of all amino acid anticodons. Of the three tRNA-coding genes without introns, one has an AT content of 76%, another 92% and the third 96%. Because of the relatively low AT, the first seems to be a possible candidate for a true gene. Its AT content was not only the lowest among the three predictions that do not have introns but also among all 64 predicted tRNA-coding genes. This is a *trnL* gene with anticodon TAA (UAA). Nevertheless, we could not confidently determine whether this gene was a false-positive prediction so we did not use it for any analyses. The only predicted *trnE* gene had an AT content of 99% and is thus very likely to be a false prediction. Therefore, the plastid genome of *Rhopalocnemis phalloides* probably lost its *trnE*, like the plastid genomes of completely heterotrophic plants from the genus *Pilostyles* (*Bellot & Renner, 2015*), although earlier *trnE* was deemed indispensable because of its function in haem synthesis (*Howe & Smith, 1991*; *Barbrook, Howe & Purton, 2006*). In the plastid genomes of *Balanophora*, the other genus of Balanophoraceae for which completely sequenced plastid genomes are available, *trnE* is present but is supposed to participate in haem synthesis only, having lost its function in translation (*Su et al., 2019*). The predicted *trnE* of *Rhopalocnemis phalloides* is located in the intergenic region between *ycf1* and *rrn23*, not where it is in *Balanophora*, which is an additional argument for the false-positive nature of this prediction. Potentially, in *Rhopalocnemis phalloides, trnE* could have been transferred to the nuclear or the mitochondrial genome, transcribed there and imported into the plastids from the cytoplasm.

Overall, the gene content of the plastid genome of *Rhopalocnemis phalloides* is similar to the gene content of the plastid genomes of *Balanophora*. All differences between them lie in differential losses of genes participating in translation, except for the loss of *trnE* in *Rhopalocnemis*, that function also in haem synthesis. Compared to *Balanophora*, *Rhopalocnemis phalloides* lacks *rps2*, *rps4*, *rps11*, *rpl14*, *trnE*, *rrn4.5*, while *Balanophora* lacks *rpl16* and *rpl36* which are present in *Rhopalocnemis phalloides*.

Most plastid genes of *Rhopalocnemis phalloides* are shorter than their homologs in close mixotrophic relatives, although not as short as homologs in *Balanophora* (Table S1). The compaction of non-coding regions in the plastid genome in *Rhopalocnemis phalloides* is also not as pronounced as in *Balanophora*, with 78.5% being coding (i.e., non-intergenic and non-intronic) in the plastid genome of *Rhopalocnemis phalloides* and 94.2% and 95.2% in the plastid genomes of *B. reflexa* and *B. laxiflora*, respectively. Several genes of *Rhopalocnemis phalloides* overlap. Namely, *ycf1* and *ycf2* overlap; so does *rpl36* which overlaps with both *rps7* and *rpl16* (Fig. 1). The intron loss in *Rhopalocnemis phalloides* is also not as pronounced as in *Balanophora*, with four cis-spliced and one trans-spliced introns remaining in *Rhopalocnemis phalloides*, whereas only the trans-spliced intron remains in *Balanophora*, with all cis-spliced introns lost.

The gene order in the plastid genome of *Rhopalocnemis phalloides* is neither collinear with the gene order of *Balanophora*, nor it is collinear with the typical gene order of photosynthetic plants (Table S2). Namely, the plastid genome of *Rhopalocnemis phalloides* has seven collinear blocks with the plastid genomes of *Balanophora* and four collinear blocks with the plastid genome of *Arabidopsis thaliana*.

## The plastid genome of *Rhopalocnemis phalloides* has a very high AT content

One of the most interesting features of the *Rhopalocnemis phalloides* plastid genome is its AT content of 86.8%. Among plant genomes, it is surpassed only by the plastid genomes of *B. reflexa* and *B. laxiflora*, two close relatives of *Rhopalocnemis phalloides*, which have an AT content of 88.4% and 87.8%, respectively (Su et al., 2019). Among prokaryotes and eukaryotes other than plants, there are also several other known genomes with higher AT content, all belonging to mitochondria or apicoplasts, with the record held by the mitochondrial genome of a fungus, *Nakaseomyces bacillisporus* CBS 7720 (Bouchier et al., 2009), with the AT content of 89.1% (according to the NCBI site, information current as of June 14, 2018).

The increased AT content is a common feature of plastid genomes of completely heterotrophic plants (Fig. 2; Table S3); to the best of our knowledge, it remains unexplained. It correlates with the degree of plastid genome reduction, with plants whose plastid genomes are the most AT rich having simultaneously some of the smallest plastid genomes.

It was the high AT content which prevented us from detecting the plastid genome of *Rhopalocnemis phalloides* from the initial assembly made of approximately 10 million read pairs. High AT content hampers PCR (Benjamini & Speed, 2012), and as library preparation for Illumina sequencing machines usually involves PCR, coverage of AT-rich

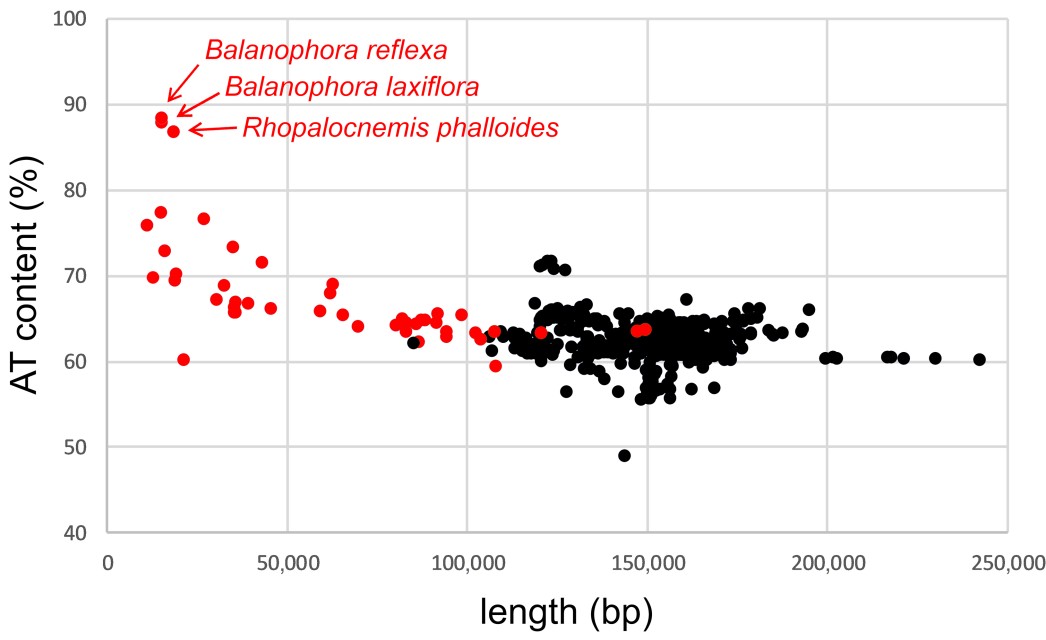

**Figure 2 AT content and lengths of the plastid genomes of Embryophyta.** Red dots denote completely heterotrophic plants and black dots mixotrophic and completely autotrophic.

regions is decreased. When we assembled the genome using an insufficient number of reads, the genome's sequence was broken into multiple contigs containing regions with relatively low AT content. The breaks occurred in the regions in which the AT content was higher; therefore, the coverage in those regions was decreased the most. The obtained sequences were not enough to determine whether the plastid genome was present because (as we describe further below) the sequences were usually similar to those from taxons other than plants, owing to the high AT content and high mutation accumulation rate. Therefore, we initially thought that these short contigs were horizontal transfers located in the mitochondrial genome. Increasing the number of reads allowed us to obtain the full sequence of the plastid genome of *Rhopalocnemis phalloides*.

The sequencing coverage in the *Rhopalocnemis phalloides* plastid genome ranges from approximately 3,000× in the least AT-rich regions to 17 in the most AT-rich regions (Fig. S1). The AT content and the sequencing coverage correlate with a Spearman's correlation coefficient of −0.93. Read insert size also depends on the AT content, with the least AT-rich regions covered by reads with an insert size of approximately 300 bp and the most AT-rich regions with an insert size of approximately 200 bp (Fig. S2); Spearman's correlation coefficient was −0.69. We suppose that the coverage drop associated with high AT content could be the reason why the authors of a work dedicated to an analysis of the *Lophophytum mirabile* (also a completely heterotrophic plant from the same family as *Rhopalocnemis phalloides*) did not observe contigs with plastid genes (*Sanchez-Puerta et al., 2017*). Additionally, in *Rafflesia lagascae*, which was reported to have no plastid genome (*Molina et al., 2014*), it may potentially be present but be unnoticed due to its high AT content. *Rafflesia lagascae* genome assembly was performed using approximately

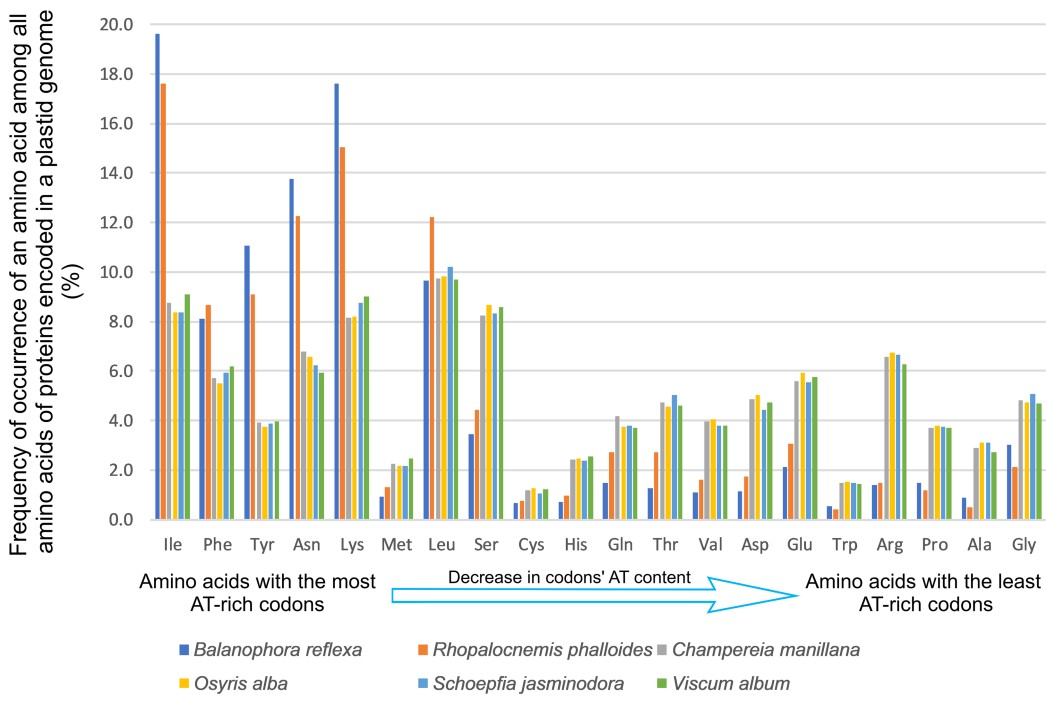

**Figure 3** Amino acid frequencies in the plastid proteins of *Rhopalocnemis phalloides* and *Balanophora reflexa* are affected by the high AT content.

400 million Illumina reads, the same amount we used for the assembly of *Rhopalocnemis phalloides*. Therefore, if *Rafflesia lagascae* indeed possesses a plastid genome, it should be much more AT-rich than the plastid genome of *Rhopalocnemis phalloides*.

The AT content is high in protein-coding genes (the average value weighted by length is 88.1%), as well as ncRNA-coding genes (the average value weighted by length is 77.5%) and non-coding regions (the average value weighted by length is 93.8%). In protein-coding genes, this led not only to a shift in codon frequencies towards AT-rich codons (Table S4) but also to a shift in amino acid frequencies in proteins, with amino acids encoded by AT-rich codons used more (Fig. 3; Table S5). For example, isoleucine, the amino acid with the most AT-rich codons, is used two times more often in the proteins encoded in the plastid genome of *Rhopalocnemis phalloides* than in homologous proteins of phylogenetically close mixotrophic species. Similarly, glycine, whose codons are among the most GC-rich, is used two times more rarely. Plastid sequences of *Balanophora* experience the same effects. Additionally, the genetic code in the plastid genomes of *Balanophora* is non-canonical, utilizing TAG (which is a stop codon in most genetic codes) as the tryptophan codon instead of the typical TGG. In contrast, the plastid genome of *Rhopalocnemis phalloides* uses TGG for tryptophan, whereas the TAG codon is not used at all, even as a stop codon.

Interestingly, such high AT content has led to convergence of gene sequences of *Rhopalocnemis phalloides* with sequences from phylogenetically distant AT-rich species. When aligning sequences of genes and proteins of *Rhopalocnemis phalloides* to sequences from NCBI NT and NR databases, respectively, the best matches are often sequences

from distantly related heterotrophic plants whose plastid genomes also have high AT content (Table S6). There are also many matches to sequences from various protists and some matches to sequences of animals and bacteria. Not all the matches are to homologous sequences, with some resulting from accidental similarity to non-coding sequences.

We thoroughly investigated one of the prominent cases of convergence—the *rrn16* gene. This particular gene was selected for the analysis because it is the only gene whose sequences are known for three genera of Balanophoraceae (*Rhopalocnemis, Corynaea, Balanophora*). This allowed us to check whether the convergence with distant species could also be observed in other genera of Balanophoraceae. BLASTN alignment of *rrn16* of *Rhopalocnemis phalloides* to NCBI NT produces two best hits to other species of Balanophoraceae, namely *Corynaea crassa* and *B. japonica* (for which only this plastid gene is sequenced and available in GenBank), whereas the next several dozen matches were to protists from the genera *Plasmodium, Nitzschia* and *Leucocytozoon*, belonging to SAR. Our initial hypothesis was a horizontal transfer from SAR to a common ancestor of the aforementioned Balanophoraceae. This was supported by the fact that a phylogenetic analysis of *rrn16* places the sequences of Balanophoraceae within SAR with very high bootstrap support (Fig. 4). A simple counterargument is that *Plasmodium, Nitzschia* and *Leucocytozoon*, though all belonging to SAR, are, in fact, quite distant phylogenetically from each other (with *Nitzschia* belonging to Stramenopiles, and *Plasmodium* and *Leucocytozoon* to Alveolata), and thus the fact that they appear in BLAST results together suggests some sort of bias. What is common for the species of the genera whose *rrn16* produces best matches to *rrn16* of *Rhopalocnemis phalloides* is that they have extremely high AT content, close to that of *Rhopalocnemis phalloides*. Among the BLAST matches, the best 20 matching sequences of SAR have an AT content ranging from 62.6% to 77.2%, with the average value 75.1%. The AT content of the *rrn16* of *Rhopalocnemis phalloides* is 77.2%. For comparison, AT contents of *rrn16* from close mixotrophic relatives of *Rhopalocnemis phalloides*, namely *V. album, O. alba, Champereia manillana and Schoepfia jasminodora*, have values typical for autotrophic plants ranging from 43.3% to 43.5%. This led us to guess that the similarity originates not from the phylogenetic relatedness of *rrn16* of *Rhopalocnemis phalloides* to *rrn16* of *Plasmodium, Nitzschia* and *Leucocytozoon* but from convergence because of their high AT content.

To test whether the grouping of *rrn16* from Balanophoraceae with SAR was a consequence of the high AT content, we removed from the multiple alignment of *rrn16* all columns that contained adenines or thymines in *Rhopalocnemis phalloides, Corynaea crassa* or *B. japonica* and rebuilt the tree three times using these three produced alignments. One may expect that if the grouping was a consequence of the high AT content, the removal of columns with adenines or thymines would lead to a relocation of the respective species from SAR. When this operation was performed for *Rhopalocnemis phalloides* (Fig. S3), it moved to a poorly resolved group containing all non-Balanophoraceae Embryophyta and also several representatives of SAR, while *Corynaea crassa* and *B. japonica* remained in the same group as in Fig. 4. When the same procedure was performed for *Corynaea crassa*, *Corynaea crassa* moved to Embryophyta, while *Rhopalocnemis phalloides* and *B. japonica* did not move (Fig. S4). Similarly, after this

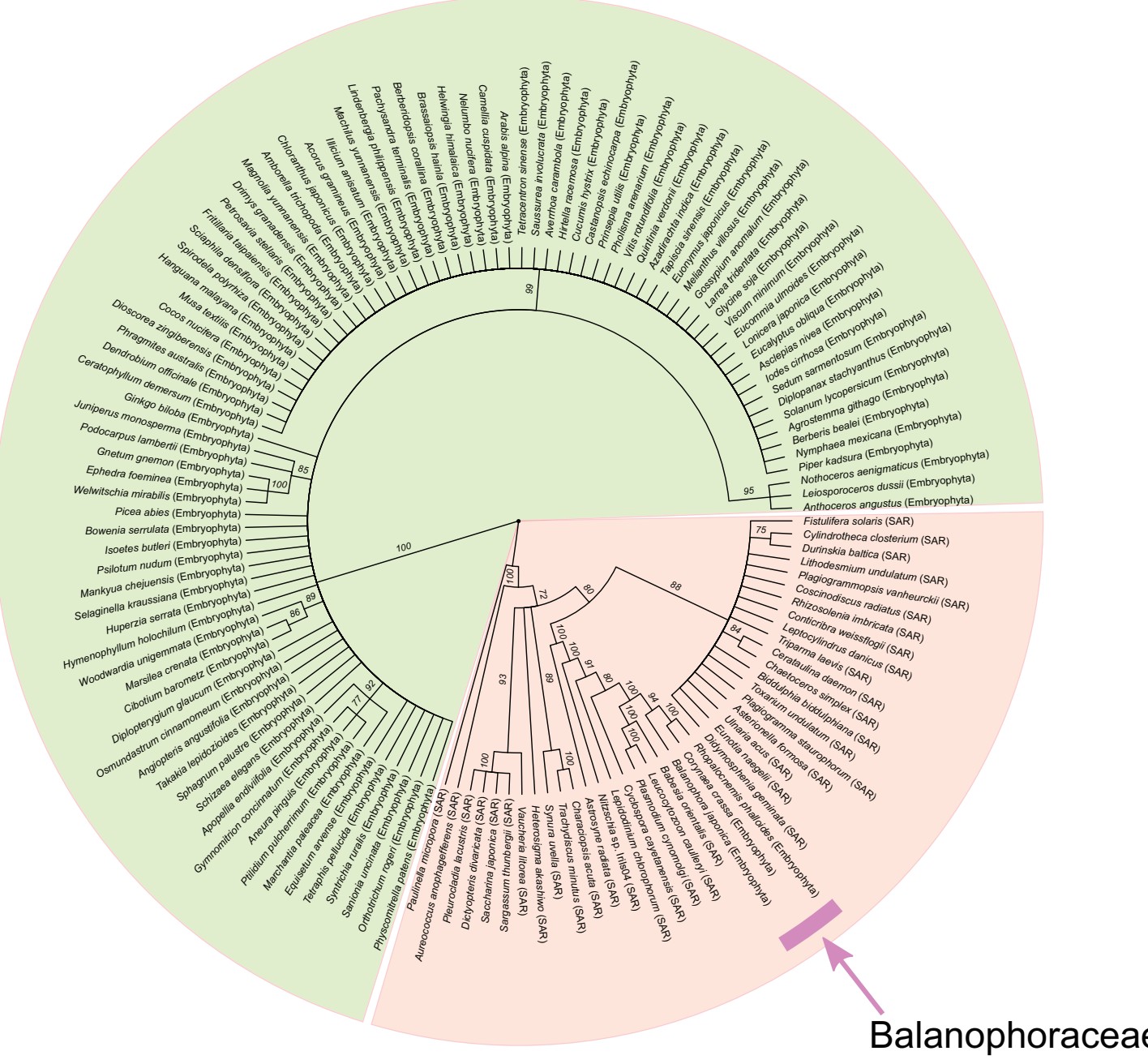

**Figure 4 The radial cladogram of *rrn16* from Embryophyta and SAR.** Branches with bootstrap support below 70 are collapsed. The group of SAR with Balanophoraceae is colored orange, while Embryophyta without Balanophoraceae are colored green. The placement of Balanophoraceae within SAR is indicated by pink.

operation was done for *B. japonica*, it moved to a group that contained all Embryophyta plus *Paulinella micropora* (a species of SAR), while *Rhopalocnemis phalloides* and *Corynaea crassa* did not (Fig. S5). Overall, although the trees are mostly poorly resolved, the fact that the removal of columns with adenines and thymines from the multiple alignment moves a species of Balanophoraceae towards Embryophyta suggests that this similarity with SAR was a consequence of AT-richness. An alternative explanation for the

seeming phylogenetic closeness of *rrn16* of these three species of Balanophoraceae to *rrn16* of SAR can be long branch attraction, but it is a characteristic problem of the maximum parsimony method and it affects phylogenetic trees built with the maximum likelihood method to a lesser degree (*Kück et al., 2012*). Additionally, the similarity of *rrn16* orthologues can potentially be a result of misalignment, but the alignment was good, and the convergence was clearly observed in the alignment (Datas S1 and S2).

Overall, our results suggest that phylogenetic analyses of heterotrophic plants (and, in general, of any species whose genomes have highly biased nucleotide composition) should be performed cautiously, as even bootstrap support values of 100 do not guarantee reliable phylogenetic reconstruction in such cases.

## Natural selection and substitution rate in the plastid genome of *Rhopalocnemis phalloides*

The nucleotide substitution rate is known to be increased in plastid genomes of heterotrophic plants, ranging from a hardly detectable increase in plants that have lost their photosynthetic ability recently (*Barrett, Wicke & Sass, 2018*) to a nearly 100-fold increase with respect to the closest photosynthetic species in the most reduced plastid genomes (*Bellot & Renner, 2015*). To the best of our knowledge, the reason for this increase is not yet known (and will be discussed in more details in the section "Why is the AT content so high?").

To compare the substitution rate in *Rhopalocnemis phalloides* with the rates in its closest mixotrophic relatives, one should first determine the phylogenetic placement of *Rhopalocnemis phalloides* relative to the species of comparison. The placement of the family Balanophoraceae has long been debated, with some scientists stating that it does not even belong to Santalales (*Kuijt, 1968*; *Cronquist, 1981*; *Takhtadzhian, 2009*). A recent work, which utilized sequences of seven genes for phylogeny evaluation, suggested that Balanophoraceae indeed belong to Santalales (*Su et al., 2015*). Moreover, the results of that work suggest polyphyly of Balanophoraceae, which consist of two clades: "Balanophoraceae A" and "Balanophoraceae B." A common feature of Balanophoraceae A is that they have highly increased substitution rates, and a common feature of Balanophoraceae B is that their substitution rates are approximately the same as in autotrophic and mixotrophic Santalales. Although it analyzed 11 species of Balanophoraceae, that study did not analyze *Rhopalocnemis phalloides*. To estimate the phylogenetic relationships of *Rhopalocnemis phalloides*, we added the sequences of its nuclear 18S rDNA and 26S rDNA to the alignment of sequences from 186 species used in that article and rebuilt the tree. As one may have expected, *Rhopalocnemis phalloides* is placed in Balanophoraceae A, with a bootstrap support value of 100 (Fig. S6). It is sister to a group of *Corynaea crassa* and *Helosis cayennensis*.

To evaluate substitution rates, dN and dS in the plastid genome of *Rhopalocnemis phalloides*, we used common protein-coding genes of this genome, the plastid genome of *B. reflexa* and plastid genomes of several other species of Santalales, available as of 2017. The genes *ycf1*, *ycf2* and *rps7* were excluded from the analysis because their sequences in *Rhopalocnemis phalloides* could not be reliably aligned with homologous sequences of

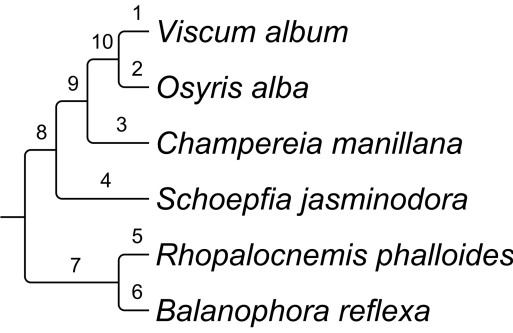

| Branch number | Substitutions per position | dN | dS | dN/dS |
|---|---|---|---|---|
| 1 | 0.069 | 0.034 | 0.219 | 0.156 |
| 2 | 0.024 | 0.010 | 0.082 | 0.128 |
| 3 | 0.034 | 0.016 | 0.108 | 0.153 |
| 4 | 0.065 | 0.024 | 0.240 | 0.098 |
| 5 | 0.747 | 0.207 | 3.067 | 0.068 |
| 6 | 0.588 | 0.179 | 2.345 | 0.076 |
| 7 | 0.593 | 0.189 | 2.326 | 0.081 |
| 8 | 0.012 | 0.014 | 0.000 | N/A[*] |
| 9 | 0.005 | 0.002 | 0.018 | 0.095 |
| 10 | 0.004 | 0.002 | 0.015 | 0.128 |

**Figure 5 Evolutionary parameters of the phylogeny of Santalales.** *Arabidopsis thaliana*, used as the outgroup, is not shown. The total length of the alignment, used for the analysis, was 3,363 bp after removal of poorly aligned regions by Gblocks. [*] dN/dS on this branch cannot be calculated owing to a very small dS value.

other species owing to the high amount of accumulated mutations. The analysis by PAML showed that the number of nucleotide substitutions in the plastid genome of *Rhopalocnemis phalloides* since the divergence from common ancestor with the mixotrophic Santalales of comparison is, on average, 21 times higher than in the plastid genomes of those mixotrophic Santalales (Fig. 5). This number should be treated with caution, as:

1. The model of nucleotide substitutions used in PAML utilizes the equilibrium codon frequencies, equal for all branches. This is definitely not the case in the studied Santalales, as the codon frequencies in the plastid genome of *Rhopalocnemis phalloides* highly differ from those in plastid genomes of mixotrophic Santalales (Table S4). We are aware of a single tool for phylogenetic analyses that can take into account different codon frequencies in different sequences. This is a program collection BppSuite. However, the analysis of these data by BppSuite provided a value of approximately 44,000 instead of 21, which was probably owing to an algorithmic mistake.

2. Non-synonymous substitutions quickly reach saturation, and thus the number of non-synonymous substitutions is underestimated for long branches (*Dos Reis & Yang, 2013*). The same is true for synonymous substitutions (*Vanneste, Van de Peer & Maere, 2013*).

3. We removed columns in the alignment with many differences between species using the program Gblocks, because such columns may result from misalignment. As regions of

genes with positive or weak negative selection accumulate mutations faster, such regions can also be potentially removed by Gblocks, leading to underestimation of substitution rates.

4. We failed to produce reliable alignments for the genes *ycf1*, *ycf2* and *rps7*, consequently the substitution rate in these genes may be higher than in others. Therefore, the exclusion of these genes from the analysis may lead to underestimation of the true substitution rate.

The substitution rate analysis for *B. reflexa* provided a very similar result to that of *Rhopalocnemis phalloides*. The dN/dS values on the branches of *Balanophoraceae* were slightly lower than on the branches of mixotrophic Santalales. Because the estimation of the dN and dS values could be imprecise, the values of dN/dS should also be treated cautiously. In the future, the problems associated with the analysis of long branches can be reduced by increasing the taxon sampling for *Balanophoraceae*, thus decreasing the branch lengths. Although the precise value of dN/dS on the branch of *Rhopalocnemis phalloides* is hard to estimate, the selection acting on its genes is definitely non-neutral, as open reading frames of all the genes are intact. If we denote the probability that there is a specific codon in a specific position as P(X), and the AT content of a gene as $\alpha$, then the probability that a random codon is a stop is

$$
\begin{aligned}
P(\text{Stop}) = P(\text{TAA}) + P(\text{TGA}) + P(\text{TAG}) &= (\alpha/2) \\
\times \ (\alpha/2) \ \times \ (\alpha/2) \ + (\alpha/2) \times ((1-\alpha)/2) \times (\alpha/2) & \\
+ \ (\alpha/2) \times (\alpha/2) \times ((1-\alpha)/2) &= \alpha^2/4 - \alpha^3/8
\end{aligned}
\tag{1}
$$

As the weighted (by length) average AT content in protein-coding genes of *Rhopalocnemis phalloides* is 88%, the probability of a random codon being a stop, as follows from this equation, is approximately 11%. This means that because stop codons are AT-rich, in a random sequence with such a high AT content as in *Rhopalocnemis phalloides*, every 9th codon will be a stop. Therefore, a strong negative selection must be acting on the genes to keep open reading frames unbroken.

## Other genomes of *Rhopalocnemis phalloides*

Sequencing of approximately 400 million paired-end reads could have been enough to assemble the mitochondrial and the nuclear genomes of *Rhopalocnemis phalloides*. The alignment by BLASTN and TBLASTN of ncRNAs and proteins, respectively, encoded in mitochondrial genomes of the reference species to the contigs of *Rhopalocnemis phalloides* revealed several dozen matching contigs with coverages of approximately 5,000× and lengths of approximately 1,000–5,000 bp. They are probably short mitochondrial chromosomes, similar to those observed in the plant *Lophophytum mirabile* (*Sanchez-Puerta et al., 2017*), also from Balanophoraceae, whose mitochondrial genome putatively consists of 54 small circular chromosomes. We are ready to provide the mitochondrial contigs upon request.

Known sizes of nuclear genomes of plants from Santalales vary from approximately 200 Mbp in *Santalum album* (*Mahesh et al., 2018*) to approximately 100 Gbp in *V. album* (*Zonneveld, 2010*). For example, if the nuclear genome size in *Rhopalocnemis phalloides* is

500 Mbp, 400 million 150-bp-long reads will produce a coverage of approximately 400 × 150/500 = 120×, which is enough for a draft assembly. To estimate the nuclear genome size, we built a k-mer frequency histogram (Fig. S7). The peak of the distribution, corresponding to the k-mer coverage of the nuclear genome, was difficult to determine, but it was below the k-mer coverage value of 2×. As the k-mer size (21) was much lower than the read size (150), the read coverage was approximately equal to the k-mer coverage. Therefore, the nuclear genome size could be estimated to be at least 400 × 150/2 = 30,000 Mbp. Potentially, the genome size could be overestimated if there is a lot of contamination (e.g., by DNA of endophytic bacteria and fungi), but a taxonomic analysis of reads suggests that contamination in unlikely to be high (Table S7). The assembly of a 30,000-Mbp-long genome is impossible using only the reads produced in the current study. Instead of the complete nuclear genome assembly, we plan to study it by means of transcriptome assembly, which is the subject of our next work.

## Why is the AT content so high?

The increase in the AT content in the plastid genomes of heterotrophic plants, as well as the increase in their substitution rates, are known and much-discussed phenomena (*Bromham, Cowman & Lanfear, 2013*; *Wicke et al., 2016*; *Hadariová et al., 2018*; *Wicke & Naumann, 2018*). However, their origin is still unknown. The simplest hypothesis for the increase in the substitution rate could be the relaxation of selection acting on genes. However, plastid genes of heterotrophic plants usually show no signs of relaxed selection, except for photosynthesis-related genes during pseudogenization. Interestingly, a high AT content and substitution rate have also been observed in plastids of non-photosynthetic protists (such as *Plasmodium*) (*Oborník et al., 2009*), which lost the genes required for photosynthesis after the transition to a heterotrophic lifestyle. Additionally, both of these phenomena have been observed in genomes of endosymbiotic bacteria (*McCutcheon & Moran, 2011*), which may be dozens of times shorter than genomes of their free-living relatives owing to the loss of genes required, for example, for biosynthesis of substances that are now provided to the symbiont by its host. Therefore, these two phenomena are probably not only unrestricted to plants but are not even related to the loss of photosynthesis.

A phenomenon that can simultaneously result in both an increase of AT content and an increase of substitution rate is the reduction in genome recombination intensity. A plastid genome is capable of recombining both within itself (the recombination of two copies of the inverted repeat) (*Zhu et al., 2016*; *Li et al., 2016*) and between two copies of a genome (*Maréchal & Brisson, 2010*). The recombination is an important step in repair, both in plastids (*Zampini et al., 2017*) and in bacteria (*Cox, 1998*), so the reduction in recombination will increase the substitution rate. Also, gene conversion in plastid (*Wu & Chaw, 2015*; *Niu et al., 2017*) as well as in bacterial (*Lassalle et al., 2015*) genomes is GC-biased, although earlier gene conversion in plastid genomes was supposed to be AT-biased (*Khakhlova & Bock, 2006*). This means that if there is a mismatch between an adenine or a thymine on one strand vs a guanine or a cytosine on the other during a recombination, it is more likely that the guanine or the cytosine will be kept, while the adenine or the thymine will be removed and replaced by a cytosine or a guanine, which is

complementary to the base on the other strand. Therefore, recombination aids in increasing the GC content in plastid and bacterial genomes, and a decrease in recombination will make a genome more AT-rich. The link between the low recombination rate and the high AT content has already been proposed for endosymbiotic bacteria with small genomes (*Lassalle et al., 2015*).

Recently, it was shown that in transcriptomes of the heterotrophic plants *Epipogium aphyllum*, *E. roseum* and *Hypopitys monotropa*, the transcript of the protein RECA1, which is required for recombination of the plastid genomes, is absent (*Schelkunov, Penin & Logacheva, 2018*). This may support the above hypothesis. However, the direct reason for the loss of RECA1 is not known. A potential explanation for the loss could be that during the transition from a mixotrophic to a heterotrophic lifestyle, plastid enzymes related to photosynthesis accumulate mutations, and since a mutated enzyme may be harmful for the organism, it is evolutionarily adaptive to accumulate the mutations very fast to quickly achieve complete disruption of a gene, instead of having a semi-degraded gene encoding a harmful protein. This effect, consisting of elimination of pseudogenes at rates faster than neutral, has already been shown to take place in bacteria (*Kuo & Ochman, 2010*). Therefore, in the period directly following the loss of photosynthesis, it may be beneficial for the plant to disturb the plastid recombination and thus disturb the repair. In fact, this process may start even before the loss of photosynthesis, because in plastid genomes of mixotrophic plants *ndh* genes often undergo pseudogenization (*Wicke et al., 2011*), and their quick removal may require an increased mutation accumulation rate. Such an increase in the mutation accumulation rate may require pseudogenization of genes of DNA replication, recombination and repair (DNA-RRR), such as *RECA1*, and once they are pseudogenised, it will be hard for a plant to return to the normal repair intensity in the plastid genome, making the transition to high mutation accumulation rates irreversible.

It is known that the mutation accumulation rate in heterotrophic plants (*Bromham, Cowman & Lanfear, 2013*), including Balanophoraceae (*Su & Hu, 2012*; *Su et al., 2015*), is also increased in nuclear and mitochondrial genomes, although to a lesser extent than in plastid genomes. These phenomena are also still unexplained. The nuclear genome contains more than a hundred (*Schelkunov, Penin & Logacheva, 2018*) genes that encode proteins, working in multisubunit complexes with proteins, encoded in the plastid genome. These are the genes encoding proteins of the electron-transfer chain, the plastid-encoded RNA polymerase, the plastid ribosome and others. When a species loses its photosynthetic ability, the nuclear-encoded genes of the electron-transfer chain are no longer under selective pressure and start to accumulate mutations. Therefore, their proteins may become harmful and may require quick elimination. Thus, the increase in the nuclear mutation accumulation rate, which may speed up the accumulation of disruptive mutations in these genes, may also by selectively beneficial. The increase in the mutation accumulation rate in the mitochondrial genome could potentially be explained by the fact that many DNA-RRR proteins are common for the plastid and the mitochondrial genomes (*Shedge et al., 2007*; *Carrie & Small, 2013*). Therefore, if it is selectively beneficial to increase the mutation accumulation rate in the plastid genome, the mitochondrial genome may also be affected.

This hypothesis of accelerated junk removal may be tested by studying plastid and nuclear genomes of many related heterotrophic species and checking whether the crumbling genes accumulate mutations at rates faster than neutral shortly after the loss of photosynthesis and whether some of the DNA-RRR genes deteriorate at the same time.

## CONCLUSIONS AND FUTURE STUDIES

The plastid genome of *Rhopalocnemis phalloides* profoundly differs from plastid genomes of typical plants, including the massive gene loss, the increased substitution rate and the high AT content. By decreasing sequencing coverage, such high AT content may "hide" plastid genomes of some heterotrophic plants, making these genomes harder to find by means of high-throughput sequencing. Alterations in the nuclear genome, accompanying these changes in the plastid genome, are an interesting issue. Our next work will be dedicated to the study of the nuclear genome of *Rhopalocnemis phalloides* by means of transcriptome sequencing.

## ACKNOWLEDGEMENTS

We are grateful to Alina Alexandrova for field assistance. We would also like to thank three reviewers: Sean Graham, Jeff Palmer and one anonymous reviewer for their valuable suggestions.

### Funding

The work was funded by a Russian Foundation for Basic Research grant (No. 16-34-01003) and a budgetary subsidy to the Institute for Information Transmission Problems (No. 0053-2019-0005). The work of Maxim Segreevich Nuraliev was carried out in accordance with a Government order for the Lomonosov Moscow State University (project No. AAAA-A16-116021660105-3). The funders had no role in study design, data collection and analysis, decision to publish, or preparation of the manuscript.

### Grant Disclosures

The following grant information was disclosed by the authors:
Russian Foundation for Basic Research: 16-34-01003.
Budgetary subsidy to the Institute for Information Transmission Problems: 0053-2019-0005.
Lomonosov Moscow State University: AAAA-A16-116021660105-3.

### Competing Interests

The authors declare that they have no competing interests.

### Author Contributions

- Mikhail I. Schelkunov conceived and designed the experiments, analyzed the data, contributed reagents/materials/analysis tools, prepared figures and/or tables, authored or reviewed drafts of the paper, approved the final draft.

- Maxim S. Nuraliev conceived and designed the experiments, performed the experiments, contributed reagents/materials/analysis tools, authored or reviewed drafts of the paper, approved the final draft.
- Maria D. Logacheva conceived and designed the experiments, performed the experiments, contributed reagents/materials/analysis tools, authored or reviewed drafts of the paper, approved the final draft.

## DNA Deposition

The following information was supplied regarding the deposition of DNA sequences:

The plastid genome is available in the GenBank database under accession MK036331. The DNA-seq and RNA-seq reads are available in the Sequence Read Archive (SRA) under accessions SRR7995544 and SRR7995545, respectively.

## Data Availability

Code of the scripts written for this study is available at Figshare:

Schelkunov, Mikhail (2019): Scripts used to study the plastid genome of *Rhopalocnemis phalloides*. figshare. Software. https://doi.org/10.6084/m9.figshare.7981496.v2.

## Supplemental Information

Supplemental information for this article can be found online at http://dx.doi.org/10.7717/peerj.7500#supplemental-information.

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
