# Peer review of "Rhopalocnemis phalloides has one of the most reduced and mutated plastid genomes known"

_PeerJ, doi:10.7717/peerj.7500_

## Round 0.1 · original submission · Major Revisions

Dear Dr. Schelkunov and colleagues:

Thanks for submitting your manuscript to PeerJ. I have now received three independent reviews of your work, and as you will see, the reviewers raised some concerns about the research, most of which are minor in context. Thus, there is plenty of all-around optimism for me to encourage you to revise your work and resubmit. I am sure that addressing these concerns will greatly improve your manuscript and bring it close to publication. Thus, I strongly encourage you to take into account all of the criticisms raised by the three reviewers.

Please note that reviewer 1 has provided a marked-up version of your resubmission.

Please note, per the comments of reviewer 2 and reviewer 3, that the Su et al. paper on Balonophoraceae plastomes should be taken into account in your revision; this may lead to additional analyses, which likely will strengthen your argument. Also, please justify the rrn16 analysis provided that a very small taxon set was analyzed in the phylogeny estimation (or delete the analysis altogether).

Please consider enlisting a native-English-speaking scientist to improve your writing and grammar (preferably one with experience in botany and comparative genomics).

I look forward to seeing your revision, and thanks again for submitting your work to PeerJ.

Good luck with your revision,

-joe

Reviewer 1 ·

Basic reporting

This paper is well-written overall. I have noted a few places on the manuscript where small wording changes might be made. I would suggest citing the recent review paper on plastome genome evolution which appears to be missing:

Graham SW, Lam VK, Merckx VS. 2017. Plastomes on the edge: the evolutionary breakdown of mycoheterotroph plastid genomes. New Phytologist 214: 48-55.

Also, in the Literature Cited, the book shown to have been written by V. Merckx was really edited by him, so individual chapters should be cited.

Experimental design

The methods are explained in good detail. Graphics are useful and informative. The report of a genome that is this small is of interest and pushes our knowledge of extreme plastomes forward.

Validity of the findings

The analyses are appropriate and the conclusions follow in straightforward manner from them. Speculative parts of the paper, such as why the genomes of these plants are so AT-rich, are identified as such.

Annotated reviews are not available for download in order to protect the identity of reviewers who chose to remain anonymous.

·

Basic reporting

This is a very interesting, well written and overall well executed study of a highly modified plastid genome in the nonphotosynthetic plant Rhopalocnemis phalloides (Balanophoraceae).

Experimental design

-- The results have not been reported on before, and this work fills in a gap in our knowledge about a poorly known group of holoparasitic plants. Like other recent reports of nonphotosynthetic plastid genomes (plastomes) in other lineages, the one described here is highly reduced through loss of photosynthesis and other genes. The plastome is unusual in other ways, including its very high AT content, as also described here. The research question is well stated.

-- The rrn16 analysis (not a huge part of the paper) is a bit odd because the taxon sampling used is really tiny. With the number of taxa included here, and the substantial distance among them, I think that most plastid genes, even of photosynthetic taxa, might behave oddly. For this reason I am not fully convinced that it fully demonstrates convergence or is the only reason for taxon attraction in the tree. The analysis might be improved with a better representation of other major land-plant lineages, for example.

-- Explain why rrn16 was chosen for phylogenetic analysis at the relevant point of the M&M (i.e. around l. 292: this was not really clear to me until later). This particular gene seems like an odd choice, as it is unusually slowly evolving in photosynthetic plants (because it is usually found in the slowly evolving IR) and so will comprise a more extreme example than other genes. This may add to the unusual results here. I am guessing the gene was used because it is one of the few genes retained in common between this organism and non-land-plants.

-- What steps were taken to make sure the plastid contigs did not include plastid inserts in the mitochondrial genome (I assume rejecting contigs that included mt genes).

-- I appreciated the extended explanation of the strategy used to find genes here (l. 171-214), which can be a difficult task in these highly modified heterotroph plastomes.

-- I think the contamination detection procedure (l. 216-235) is most relevant to the discussion of the nuclear genome (e.g. 536, Table S5), but this was unclear to me until I had read the whole paper. A brief explanation of the purpose of this procedure should be included in the M&M section.

Validity of the findings

-- Su et al. (PNAS doi: 1816822116) recently reported retention of trnE genes in other Balanophoraceae species that are likely non-functional in protein synthesis, but could still be functional for their alternative role in haem-biosynthesis. Could that be the case here? The tRNA analysis and/or discussion (particularly for this gene) could be re-done in light of this.

-- I also wondered if the ‘false-positive’ trnE reported here is located close to one or more genes it is found in for photosynthetic organisms (or other Balanophoraceae). More generally, how does plastid genome structure here compare to other photosynthetic or non-photosynthetic relatives (e.g., Su et al. 2018)? How about including a brief Mauve analysis to summarize this?

-- I appreciated that caveats to estimating dN/dS ratios were laid out in the Results/Discussion (l. 479-503). The logic of why genes with open reading frames in a lineage with high AT content are likely to still be under strong negative selection (estimation of no. of stop codons if random) is a simple but effective way of thinking about this.

Additional comments

Overall I thought this paper was well written and said thought-provoking things.

I’d like to see the findings reported here put into context with recently published findings for two other Balanophoraceae plastomes (Su et al. PNAS doi: 1816822116) in terms of things like AT content, gene content, unusual trnE presence/possible function, novel genetic code, genome structure, lack of inverted repeat, and other findings reported there.

I think it could be stated more directly what the authors think was the main reason for success in eventually obtaining the plastid genome after initial failure to do so. Was the main breakthrough just the substantial increase in depth of coverage (l. 94), the complex strategy used to recover plastid genes (l. 271-214), a combination of these… or something else? This could be relevant for other purported losses (e.g., Molina et al. 2014, reported for Rafflesia), or for other uncooperative taxa.

I am not sure I buy the rationale for why the AT content is so high (i.e., linking this to possible loss of nuclear recombination genes, with that favoured as a mechanism to rapidly degrade plastid genes that have lost function but are still translated and so potentially interfere with other processes, presumably). A high mutation rate (in plastid or other genomes) could have many very undesirable pleiotropic effects – would these really outweigh the benefits from getting rid of a few photosynthesis genes? However, this speculation is still worth including, and I like the suggestion to look for loss of DNA-RRR genes in other heterotroph nuclear genomes.

I understand that the authors may not want to release the raw sequencing reads yet if working on other aspects of it. The plastid genome seems to have an NCBI number (MK036331, noted in Table S1), although this could usefully be stated in the main text too. Perhaps the mitochondrial contigs could also be released here, instead of by request, as the authors state they won’t work further on these.


Minor points

Para starting l. 46 – gene loss and retention (e.g., of the five genes mentioned here) was also reviewed in Graham et al. (2017 New Phytol 214:48-55), which could be cited here.

l.23. ‘outstanding’ used in the Abstract is unusual English usage. Perhaps use something like ‘extraordinarily high’ instead.

l. 34. it is not clear that full mycoheterotrophs are fully parasitic. Workers in the field are usually cautious about claiming so; I believe the thinking is that perhaps the fungi get something other than photosynthate from the symbiosis. It is hard to disprove that they don’t of course, but mycoheterotrophs clearly started off as mutualistic, while holoparasites did not. So there may be something to this objection.

l. 70. Make clearer here that there must be relaxation of selection in these genomes (explaining photosynthetic gene loss), although that doesn’t enter into this argument much.

l. 83. It would be safer to call this a *probable* loss of a plastome in Rafflesia, as perhaps one could be recovered with additional effort (as was done here, and later suggested for Rafflesia, l. 393).

l. 153-154. Briefly state/explain the problem noted by Shah et al. (2018).

l. 297. ‘SAR’ should be explained on first usage (not just spelled out later in the same paragraph).

l. 330-331. The speculation that RNA components of the plastid ribosome (rDNA gene products) may be imported was raised in Graham et al. (2017 New Phytol 214:48-55), and has been speculated on for quite a while for tRNA genes (e.g., Wolfe et al. 1992).

Fig. 4 (and S2). I find these trees hard to read because of the very dark colours used behind taxon names. I am also red-green colour blind, so Fig. 2 is the worst possible colour scheme for me and a large fraction of the population.

l. 406 and elsewhere. Avoid ‘interestingly’ (let the reader decide what is interesting)

l.467. Remind reader than only two genes of R. phalloides were included here. The total number of genes used for Fig. S5 could also be noted in the figure legend there.

l. 503. You could usefully note that (much) better taxon sampling could help with dN/dS estimation here.

l. 581. ‘semi-crumbled’ is evocative, but ‘semi-degraded’ might sound more scientific in English?

·

Basic reporting

The introduction and background are well done, with relevant issues presented and appropriately referenced.

The structure of the paper conforms to disciplinary and Peer J standards.

As far as the raw data being supplied, if what is meant here is that, in the case of this particular paper, appropriate GenBank and SRA accession numbers were provided, then yes, the raw data were supplied.

The writing uses reasonably professional English and is certainly better than most papers I see that are written by non-native English speakers. However, there are a number of places where the writing could be improved (e.g., rewrite the phrase “many times discussed phenomena” on page 19) or where an inappropriate word is used (e.g., also on p. 19, change “reparation” to “repair”, specifically “DNA repair”). If possible, the authors should enlist a native-English-speaking scientist who is reasonably close in field to this paper to give it a quick read for grammar and word choice.

Three of the main figures should be improved as follows:

Fig. 1. Font sizes should be increased significantly throughout the figure. The colored arrows indicating genes should heavier/thicker so that the color differentiation is more apparent. The internal (boxed) legend should either be deleted (and this information provided in the regular legend) or moved below the actual figure so that the figure can be larger and more readable. In this figure, in other figures, and in the text, numbers larger than 999 should include commas (as in “2,000”) rather than a space (as in “2 000”).

Fig. 4. Please reduce the intensity/darkness of the three colors used so that the black lettering is more easily read.

Fig. 5. The current format makes it very hard to readily compare the various values on the branches. I think this figure would be much improved if it were recrafted so that the cladogram is shrunken greatly in width, with the only numbers on it being 1-8 (one number on each of the eight branches), and with the d, dn, etc. values placed in a table underneath the figure, with the 1-8 numbers being the first column in the table.

Experimental design

The experimental design is fine as is, with the Materials and Methods section being appropriately thorough, detailed, and informative. This study represents a rigorous investigation.

Validity of the findings

The data in this paper are robust, and the conclusions are generally well stated and drawn – in the context of the relevant literature available to the authors at the time of manuscript submission – and are relevant to the questions and results of this study.

There is, however, a big issue here. On Dec. 31, 2018, subsequent to submission of the current manuscript, a very highly relevant paper was published online in PNAS with the title “Novel genetic code and record-setting AT-richness in the highly reduced plastid genome of the holoparasitic plant Balanophora”. (Note that this I was a coauthor on this PNAS paper.) It is imperative that the current MS be thoroughly revised to include all relevant comparisons to and discussion about the Balanophora findings, this being a member of the same family of parasitic plants as Rhopalocnemis, and with the two sets of plastid genomes sharing a number of interesting, derived features – most notably with respect to high AT content – yet also differing in one or two important ways. Among these revisions are:

A) Please point out wherever appropriate, starting with the abstract, that the two sequenced Balanophora plastid genomes are slightly more AT rich than the R.p. genome, and, naturally enough, have very similar codon and protein usage biases.

B) The authors should consider comparing the AT richness and lengths of those genes that are held in common by the R.p. and Balanophora genomes, as it evident from both papers that there is considerable variation in AT-richness on a gene by gene basis, while the Balanophora paper also reports a significant reduction in size for a number of its protein genes, with ycf2 being exceptionally reduced in size (but apparently less reduced in R.p.).

C) Both Balanophora genomes should be included in Fig. 2.

D) It would be best to include at least Balanophora genome in Fig. 3.

E) One, and possibly both, Balanophora genomes should be included in Fig. 5.

F) The authors should carefully consider whether the Balanophora genomes should be included in any of the supplementary figures and tables.

G) Please compare the inferred gene content between the two genomes. As annotated, the R.p. genome contains two genes (rpl16 and rpl36) that appear to be absent from the Balanophora genomes. Vice-versa, the Balanophora genomes contain four protein genes (rps2, rps4, rps11, rpl14), one tRNA gene (trnE), and one rRNA gene (rrn4.5S) that appear to be absent from the R.p. genome. I think it is incumbent on the authors to investigate even more closely than they already have done the possibility that they have failed to detect one or more of these six genes (as part of this they may of course choose to see whether, conversely, the Balanophora study missed rpl16 or rpl36).

H) Somewhat related to the last point, the authors should calculate how much of R.p. genome is non-genic (i.e., not exons or introns, including, as done in the Balanophora paper, excluding a guesstimate for the length of the trans-spliced intron of rps12) and compare this value to those of the Balanophora genomes. The Balanophora paper makes a big point about how compact the the two Balanophora genomes are, while Fig. 1 suggests that the R.p. genome is a good bit less compact. On the other hand, the larger spacer regions in R.p. do raise the possibility that the authors, despite the care and thoroughness with which they’ve already addressed this issue, somehow failed to detect one or more plastid genes.

I) Assuming, as is probably the case, that R.p. does in fact lack the trnE gene, then this is a particularly important gene content different relative to Balanophora that deserves a bit of discussion, especially given that the Balanophora trnE gene is proposed to function only in heme synthesis, but not in protein synthesis.

J) The authors should consider pointing out one notable difference between these genomes, which is that the Balanophora genomes have lost all cis-spliced introns, whereas the R.p. genome contains 4 cis-spliced introns, all of them located in genes that are present in the Balanophora genomes.

K) The authors MUST discuss the most important apparent difference between these genomes, which is that the Balanophora genomes use a novel genetic code, in which TAG has been reassigned from stop to tryptophan, whereas implicit in the fact that there is no mention of the genetic code in the current manuscript is that R.p. presumably uses the standard genetic code. As part of this, the authors should carefully examined their alignments and annotations to see if there’s any possibility that R.p. might share with Balanophora the TAG code deviation. Is it possible that there is only a few, even just one, amino-acid position in an R.p. protein gene in which TAG appears in-frame at a site that is normally TGG/trptophan in other plants? Related, could some of the current annotations be extended with respect to homology to other plants by reassigning an in-frame TAG from stop to Trp? One possibility that should be considered in assessing a possible code change in R.p. is that in contrast to Balanophora, where the genetic-code change is “complete” (i.e., TAG is used only for Trp, i.e., never as a stop codon), it could be “incomplete” in R.p. That is, is it possible that R.p. uses TAG for BOTH Trp and Stop. Relevant here are references 56 and 57 in the Balanophora paper, which report cases where two unrelated green algal plastid genomes use TGA in this manner, i.e., as both stop and sense codons.

Here are a few suggestions, mostly minor, for improving the text as currently written:

The last two sentence of the brief Conclusions section are not conclusions but constitute a statement about an intended follow-up study on the nuclear genome/transcriptome of R. phalloides. One solution here is to change the title of the section to “Conclusions and future studies”. On the other hand, the nuclear study is just one of many possible future/follow-up studies and it is already mentioned in lines 537-539. So it might be better to exclude the last two sentences of the Conclusion section and beef up the actual conclusions part.

Abstract: Please change “Their genomes, especially plastid genomes, markedly differ…” to “Their plastid genomes markedly differ…” as I don’t think one can safely generalize int that the mitochondrial and nuclear genomes of heterotropic plants “differ” (much less “markedly") from those of photosynthetic plants.

Lines 57-58: I suggest qualifying “ycf1 (encodes a component of the translocon…) to something like “ycf1 (though to encode a component of the translocon).

Line 83: Please point out that in the case of Rafflesia, it’s possible that the authors of the Rafflesia study failed to detect a highly divergent plastid genome, a possibility raised in both the Hydnora and Pilostyles papers, as well as by the authors of the current studies themselves, later on in the manuscript.

Line 322 and in the legend to Fig. 1, it’s important to qualify/change “circular” to “circular-mapping”.

Additional comments

Overall this is a well done, thoughtful, and interesting study.

---

## Round 0.2 · Minor Revisions

Dear Dr. Schelkunov and colleagues:

Thanks for revising your manuscript to PeerJ. I have now received one independent review of your work, and as you will see, there are still a few minor issues to address. Once these issues are handled please resubmit your work. I am sure it will then be ready for publication.

I look forward to seeing your revision, and thanks again for submitting your work to PeerJ.

-joe

·

Basic reporting

See below

Experimental design

See below

Validity of the findings

See below

Additional comments

The revised MS is nicely improved, with appropriate inclusion of (in text, figures and tables) and comparision to the recently published Balanophora plastome sequences, as requested by myself and reviewer.

My only substantial concern is with Fig. 4, where I agree with reviewer 2’s point about the tiny taxon sampling (I feel quite amiss that I didn’t raise this point myself in my review) and disagree with the author’s rebutta. To do a meaningful phylogenetic analysis of this issue requires far more comprehensive taxon sampling, not only of other land-plant lineages but also a thorough sampling of the SAR group. To include only the best BLAST hits to Rhopalocnemis seems almost preordained to lead to the kind of result shown in Fig. 4B. Keeping the figure as is will only detract from an otherwise fine study. If the authors are not prepared to do a proper sampling of taxa than I think it would be in their own best interests to delete this figure and analysis, and to just let the BLAST results stand on their own as evidence of potential convergence due to high AT-content. In this respect, the authors might consider stating in the text the AT content of the 16S rRNA genes of Nitschiza, Plasmodium, etc.

Line 449: “is supposed to be” unnecessarily (and I’m sure unintentionally) casts suspicion, doubt on the conclusion of Su et al. that Balanophora uses a non-canonical code. Please reword, preferably to “is” as in “…the genetic code in the plastid genomes of Balanophora is non-canonical….”

---

## Round 0.3 · accepted · Accept

Dear Dr. Schelkunov and colleagues:

Thanks for revising your manuscript to PeerJ, and for addressing the concerns raised by the reviewers. I now believe that your manuscript is suitable for publication. Congratulations! I look forward to seeing this work in print, and I anticipate it being an important resource for research communities studying the Rhopalocnemis phalloides and overall evolution of plastid genomes.

Thanks again for choosing PeerJ to publish such important work.

-joe